# The Vulvar Cancer Risk in Differentiated Vulvar Intraepithelial Neoplasia: A Systematic Review

**DOI:** 10.3390/cancers13246170

**Published:** 2021-12-07

**Authors:** Féline O. Voss, Nikki B. Thuijs, Ravi F. M. Vermeulen, Erica A. Wilthagen, Marc van Beurden, Maaike C. G. Bleeker

**Affiliations:** 1Department of Pathology, Cancer Center Amsterdam, Vrije Universiteit Amsterdam, Amsterdam UMC, 1081 HV Amsterdam, The Netherlands; f.o.voss@amsterdamumc.nl (F.O.V.); n.thuijs@amsterdamumc.nl (N.B.T.); 2Department of Gynecology, Antoni van Leeuwenhoek Hospital, The Netherlands Cancer Institute, 1066 CX Amsterdam, The Netherlands; r.f.m.vermeulen@amsterdamumc.nl (R.F.M.V.); m.v.beurden@nki.nl (M.v.B.); 3Scientific Information Service, Antoni van Leeuwenhoek Hospital, The Netherlands Cancer Institute, 1066 CX Amsterdam, The Netherlands; e.wilthagen@nki.nl

**Keywords:** differentiated vulvar intraepithelial neoplasia, dVIN, vulvar squamous cell carcinoma, absolute risk, recurrent risk, natural course

## Abstract

**Simple Summary:**

We aimed to systematically review current literature on the risk of developing vulvar squamous cell carcinoma in patients with differentiated vulvar intraepithelial neoplasia. Amongst clinicians, this latter entity is known as a rare premalignant vulvar lesion, which is often found adjacent to vulvar cancer. Knowledge on the exact cancer risk in differentiated vulvar intraepithelial neoplasia can help guide clinicians in the treatment and surveillance of these patients. This review confirms that patients with differentiated vulvar intraepithelial neoplasia have a high risk of developing vulvar squamous cell carcinoma and that cancer progression can occur rapidly. The results of this study highlight the importance of clinical awareness and prompt identification of differentiated vulvar intraepithelial neoplasia, given its high malignant potential. Therefore, further research on this disease should be encouraged.

**Abstract:**

Differentiated vulvar intraepithelial neoplasia (dVIN) is the precursor of human papillomavirus (HPV)-independent vulvar squamous cell carcinoma (VSCC). Given the rare incidence of dVIN, limited information on the exact cancer risk is available. We systematically reviewed the primary and recurrent VSCC risk in patients with dVIN, as well as the time to cancer development. A systematic search was performed up to July 2021 according to the PRISMA guidelines. Five reviewers independently screened articles on title, abstract and full text, followed by critical appraisal of selected articles using the Quality in Prognostic Studies (QUIPS) tool. Of the 455 screened articles, 7 were included for analysis. The absolute risk for primary VSCC in dVIN varied between 33 and 86%, with a median time to progression to VSCC of 9–23 months. The risk of developing recurrent VSCC in dVIN associated VSCC was 32–94%, with a median time to recurrence of 13–32 months. In conclusion, patients with dVIN have a high risk of developing primary and recurrent VSCC with a short time to cancer progression. Increased awareness, timely recognition, aggressive treatment and close follow-up of HPV-independent vulvar conditions including dVIN is therefore strongly recommended.

## 1. Introduction

Vulvar intraepithelial neoplasia (VIN) is the precursor lesion of vulvar squamous cell carcinoma (VSCC), which is categorized into human papillomavirus (HPV)-induced vulvar high-grade squamous intra-epithelial lesions (vHSIL) and HPV-independent differentiated VIN (dVIN) [1,2,3,4,5]. The vast majority of high-grade VIN lesions are diagnosed as vHSIL, with a known cancer risk of 3–10% [6,7,8,9]. DVIN comprises less than 5% of VIN lesions [10]. Nevertheless, the majority of VSCC are HPV-independent, indicating the high malignant potential of dVIN [1,2,5].

After the first dVIN case was described by Abell and Gosling in 1961 as an ‘intraepithelial carcinoma of simplex type’, it was not until 1986 that the terminology ‘differentiated VIN’ was adopted by the International Society for the Study of Vulvovaginal Disease (ISSVD) [11,12]. In 2004 the ISSVD proposed a classification scheme distinguishing HPV-induced high-grade vulvar neoplasia, formerly known as usual type VIN (nowadays vHSIL), from HPV-independent dVIN [13].

Besides the absence of HPV infection, dVIN has several other clinical and pathologic features that distinguish it from the more frequently diagnosed vHSIL. DVIN almost always arises in a background of lichen sclerosus (LS), a chronic inflammatory dermatosis which mainly occurs in the anogenital area of postmenopausal women. Both LS and dVIN typically occur in women above the age of 60 years, although it can affect younger women as well [14].

Although dVIN represents the minority of diagnosed VIN lesions, it is often diagnosed adjacent to VSCC [8]. DVIN is therefore regarded as the more aggressive precursor lesion when compared to vHSIL and other non-neoplastic epithelial disorders, such as LS and lichen planus [14]. Given the aggressive nature of dVIN, it is currently receiving increasing attention concerning methods to improve diagnostics and clinical management. Solitary dVIN is a rare finding and difficult to diagnose due to its varying presentation. Therefore, limited information on the exact cancer risk is available. As a result, current treatment of dVIN, consisting of surgical excision, is based on small case series and expert opinions.

The purpose of the present study was to review current literature on the risk of developing primary and recurrent VSCC in patients with dVIN, including the time to cancer progression. Better insight in the cancer risk can create a more scientific basis for an evidence-based guideline for the treatment of dVIN.

## 2. Materials and Methods

The methods and results are reported according to the Preferred Reporting Items for Systematic Reviews and Meta-Analyses (PRISMA) statement [15]. This study was registered with Research Registry and the unique identifying number is: reviewregistry 1243 (https://www.researchregistry.com, accessed on 1 November 2021).

### 2.1. Search Strategy

To identify all relevant publications on cancer risk in dVIN, we performed a systematic electronic search in bibliographic databases Medline, Embase (Ovid) and Scopus up to 13 July 2021. The following terms, including synonyms and closely related words, were used as index terms and free-text words: “differentiated or HPV-independent” and “vulvar intraepithelial neoplasia” or “dVIN” (Appendix A). No limits were applied, and duplicate articles were excluded using EndNote.

### 2.2. Eligibility Criteria and Study Selection

Five medical doctors and experts in vulvar pathology (MCGB, MvB, RFMV, NBT, FOV) independently screened all titles and abstracts for eligibility according to predetermined in- and exclusion criteria, using Rayyan QCRI. This was followed by independent full-text screening of the selected articles. Articles were regarded as eligible to be included whenever one or both of the following aspects were assessed: VSCC risk in patients with dVIN and/or the risk of recurrent VSCC when arising in a background of dVIN. Case reports and articles of which full text was unavailable were excluded. Studies were also excluded if no separate analyses had been performed for high-grade VIN lesions to discriminate between vHSIL and dVIN. Reference lists of included articles were cross-checked manually to identify any additional studies. Selected articles were discussed in a consensus meeting with all reviewers to reach a final selection.

### 2.3. Data Extraction and Analysis

Data from selected studies were extracted from the full text or tables by four reviewers (MCGB, RFMV, NBT, FOV). Data extraction included first author, year of publication, country of investigation, study design, type of cohort, inclusion period and criteria, number and type of cases included (dVIN without history of VSCC and/or dVIN adjacent to VSCC), age at dVIN and/or dVIN adjacent to VSCC diagnosis, follow-up time, number of primary and/or recurrent VSCC cases and time to progression to (recurrent) VSCC. The absolute risk of primary and/or recurrent VSCC development was extracted or calculated if adequate data were provided. Absolute risk was calculated by dividing the number of cases developing primary and/or recurrent VSCC by the number of dVIN and/or dVIN adjacent to VSCC cases. Data were analyzed in a descriptive manner if meta-analysis could not be performed. The data were tabulated in Microsoft Excel version 2016 (Microsoft Corporation, Redmond, WA, USA).

### 2.4. Critical Appraisal

The quality and risk of bias of included studies were independently assessed by two reviewers (NBT, FOV) using the Quality in Prognostic Studies (QUIPS) tool of the Cochrane Prognosis Methods Group [16]. The QUIPS tool includes the following six bias domains: study participation, study attrition, prognostic factor measurement, outcome measurement, study confounding, and statistical analysis and reporting. Studies were evaluated for their reliability and eligibility for each of the six domains, rating as low, moderate or high potential risk of bias. Discrepancies were resolved in a consensus meeting.

## 3. Results

### 3.1. Search Results

The systematic literature search and selection process is outlined in the PRISMA flow diagram (Figure 1) [15]. A total of 901 articles were found in Medline, Embase and Scopus databases using our selected search. After removal of duplicates, 455 articles were selected for primary screening, after which 418 articles were excluded based on title and abstract. Of the 37 articles sought for retrieval, full-text reports were not available of 7 articles. After full-text assessment, an additional 23 articles were excluded which did not meet our inclusion criteria. This selection procedure resulted in seven articles to be included for analysis.

### 3.2. Study Characteristics

The characteristics of all seven included studies are summarized in Table 1. Three studies were carried out in the Netherlands by three different institutes [7,17,18], of which two were population-based studies and one was a center-based study. The other four studies were center-based studies carried out in Austria [19], Canada [20], the UK [21] and the USA [22]. All had a retrospective cohort study design. The number of included dVIN cases varied from 7 to 197. The study periods in which dVIN cases were included varied from 5 to 20 years, with cases included between 1985 and 2016. From one study, data on the inclusion period was not available.

### 3.3. Risk of Bias of All Included Studies

Selected studies underwent quality assessment according to the QUIPS tool (Table 2). The more recent published studies had an overall lower risk of potential bias for all domains. All of the studies scored a moderate to high potential risk of bias concerning the study confounding domain. This was the most common methodological weakness, as important confounders relevant to primary and recurrent VSCC risk, such as treatment of primary dVIN or pathologic free margins in dVIN associated VSCC, were often not taken into account or measured inadequately.

### 3.4. Outcome of Objectives

#### 3.4.1. Primary VSCC Risk in dVIN

Five studies, including two population-based and three center-based studies, studied the risk of developing primary VSCC in dVIN patients (Table 3) [7,18,19,20,22]. Sample sizes varied from 7 to 67 dVIN patients, with age ranging from 67 to 75 years. The absolute cancer risk in women with dVIN was calculated with the total number of VSCC cases in each separate dVIN cohort and ranged between 33 and 86% across the five included studies.

The time to primary VSCC progression was investigated in four of the five studies [7,18,20,22]. The median interval between dVIN diagnosis and VSCC diagnosis ranged from 9 to 23 months. Pooling of results was not possible because two studies did not report any information regarding follow-up time [18,19], and the other three studies reported differences in follow up time, ranging from 72 to 167 months [7,20,22]. No correlation was found between the duration of follow-up and the cancer risk observed.

#### 3.4.2. Recurrent VSCC Risk in dVIN

Three center-based studies investigated the risk of recurrent vulvar cancer in patients treated for dVIN associated VSCC (Table 3) [17,20,21]. Sample sizes varied from 18 to 197 patients with dVIN adjacent to VSCC, with age ranging from 73 to 76 years. Overall, the recurrence risk ranged between 32 and 94%. The median time to recurrence was reported in the two studies with the highest VSCC recurrence risk and was 13 and 32 months [17,20].

Higher local VSCC recurrence rates were observed when dVIN was specifically located in the resection margins compared to patients with dVIN found adjacent to the tumor but not in the resection margins (61% versus 42% after 10 years, respectively, *p* = 0.002) [17]. The study which reported the highest recurrent cancer risk (94%) reported that 7/20 (35%) and 15/20 (75%) surgical specimens had positive resection margins for invasive carcinoma or dVIN, respectively [20]. All patients with positive margins for VSCC received adjuvant radiotherapy in this study.

## 4. Discussion

This is the first systematic review summarizing the primary and recurrent risk of VSCC in women with dVIN, including the time to primary and recurrent VSCC. Patients with dVIN have a high risk of developing VSCC, with a reported absolute primary VSCC risk of 33–86% [7,18,19,20,22] and a recurrent VSCC risk of 32–94% [17,20,21]. The median time to primary vulvar cancer progression varies widely, but nearly all cases occurred within 2 years after dVIN diagnosis [7,18,20,22]. VSCC recurrences after primary VSCC with adjacent dVIN occurred mostly within 3 years after primary vulvar cancer diagnosis [17,20].

There are a number of limitations of this review. First of all, only seven studies met our inclusion criteria with high heterogeneity between the studies. In contrary to the broad consensus that dVIN is a condition with a high cancer risk, the low number of studies on this subject is illustrative of the relatively limited amount of research that has been done on dVIN. Moreover, these studies are difficult to conduct as solitary dVIN is a rare diagnosis. Due to rapid progression to carcinoma, dVIN is often diagnosed adjacent to VSCC, explaining the higher sample sizes observed in studies on recurrent cancer risk compared to those studying the primary cancer risk.

A second limitation is the selection bias observed in a number of studies. Interestingly, the study with the largest cohort reported the lowest absolute primary vulvar cancer risk, which is a population-based study [18], whereas the highest primary cancer risk was found in the study with the smallest cohort, which is a center-based study [20]. Nearly all center-based studies on solitary dVIN have relatively small sample sizes. Although center-based cohorts are more prone to selection bias, most of these studies did revise the pathology slides to confirm dVIN diagnosis, leading to a less biased selection [17,19,20,22].

In contrast, the two population-based studies contain more cases of dVIN and lack selection bias [7,18]. Both studies observed that dVIN occurs more frequently in older women, with a peak incidence amongst women between age 75 and 85 years. Although these are more representative cohorts of the general population, histopathologic review of cases was not performed. Moreover, the two studies used different definitions of dVIN when selecting cases. In addition to ‘differentiated VIN’ and ‘VIN simplex type’, Van de Nieuwenhof et al. also included cases with ‘VIN NOS (not otherwise specified) with LS and/or an HPV negative result’ in the dVIN group, while Thuijs et al. did not include these latter cases. This seems to influence not only the number of included dVIN cases (67 vs. 12, respectively), but also the observed primary cancer risk (33% vs. 58%, respectively). This is further supported by a subgroup analysis performed by Thuijs et al., showing a 10-year cumulative VSCC incidence of 38% in 43 patients who had both vHSIL and LS, which is comparable to the cancer risk in the dVIN population reported by van de Nieuwenhof et al. [7,18].

Another limitation of this review is the retrospective design of all studies. This most likely resulted in the lack of information on relevant confounders, such as age, treatment, comorbidities and medical history. An important confounder which was rarely reported is the effect of dVIN treatment on the primary cancer risk. As spontaneous regression of dVIN is unlikely to occur, patients with dVIN are surgically treated by wide local excision with the aim of free resection margins [23]. Needless to say, dVIN patients treated by radical excision will probably have a lower risk of subsequent malignancy. Only one study reported on surgical treatment in patients with dVIN without a history of VSCC [22]. In this study, the three patients who developed VSCC only had a diagnostic biopsy taken, whereas the other patients who did not develop VSCC underwent total or partial vulvectomy or an excision of the dVIN lesion. This emphasizes the importance of radical excision of dVIN to prevent malignant progression.

Nevertheless, even after treatment, recurrent dVIN lesions are common as the remaining anogenital area adjacent to the removed dVIN lesion is often abnormal or affected with LS. Hence, close and lifelong follow-up as well as adequate treatment of both LS and dVIN is needed. The primary treatment of LS consists of maintenance therapy with ultra-potent topical corticosteroids (UTCS), which has shown to prevent the progression towards VSCC [24]. As dVIN mostly arises in a background of LS, UTCS maintenance treatment after surgical excision of dVIN should be considered to prevent recurrent dVIN and subsequent VSCC.

Finally, the studies on recurrent VSCC risk reported a wide variance in the resection margin status of the surgical specimens. In the study with the highest vulvar cancer recurrence rate amongst patients treated for dVIN associated VSCC, 35% of primary surgical specimens had positive margins for invasive carcinoma [20]. This inherently influences the chance for recurrent vulvar cancer, which therefore must be taken into account when interpreting this result. This also applies to resection margins being positive for dVIN. Although only one study investigated this, the presence of dVIN in the resection margin leads to a higher risk of recurrent cancer compared to patients without positive margins for dVIN (10-year local recurrence risk of 61% vs. 42%, respectively) [17]. This implies that not only the vulvar cancer should be radically excised, but also all the adjacent dVIN to prevent recurrent disease. Current guidelines do not give clear recommendations on what to do when dVIN is present in the pathologic resection margin after VSCC excision [25,26,27]. Consequently, current clinical practice with regards to positive dVIN margins depends on the expert opinion of the treating gynecologic oncologist. However, given the increased recurrent VSCC risk in patients with positive resection margins for dVIN, it is reasonable to consider re-excision of surgical scars in such cases.

Diagnosing dVIN is challenging for clinicians, both histologically and clinically. The histopathologic morphology may vary considerably. In recent years, two new HPV-independent precursors of VSCC have been identified, termed differentiated exophytic vulvar intraepithelial lesion (DE-VIL) and vulvar acanthosis with altered differentiation (VAAD) [28,29,30]. These entities show overlapping morphology with dVIN, and previous studies suggest that a molecular association exists with verrucous carcinoma, a rare variant of HPV-independent VSCC [29,31]. However, many of the histological features, such as parakeratosis, absence of a granular layer, premature keratinization and nuclear atypia, are not specific for dVIN, DE-VIL and VAAD only [32,33,34]. HPV-independent precursors of VSCC are therefore easily mistaken for inflammatory atypia, squamous hyperplasia or other non-neoplastic epithelial disorders. A study in 2011 found that 42% of LS cases were reclassified as dVIN lesions after revision by a specialized gynecologic pathologist [35]. Yet, even for experienced pathologists with gynecological expertise the diagnosis remains challenging. A study evaluating the reproducibility of the histopathologic diagnosis of dVIN uncovered a low interobserver agreement between pathologists, which did improve in gynecologic pathologists after specific guidelines with histological characteristics were provided [36]. More recently, a survey amongst pathologists uncovered that basal layer atypia was the only essential feature of dVIN over which consensus was reached [34]. Thus, the histopathological diagnosis of dVIN and other HPV-independent precursors of VSCC can be subtle and prone to misdiagnosis.

Likewise, clinical recognition of dVIN is difficult due to its varying presentation. It often presents as a focal grey-white or red colored roughened surface, but may also appear as a thickened white plaque, wart-like or as an atrophic or ulcerative lesion [10,37]. After surgical treatment of VSCC, it becomes even more difficult to clinically visualize a dVIN lesion. Surgical scarring can mimic LS and vulvar neoplasia, resulting in a possible residue or recurrent dVIN to be easily mistaken for fibrotic tissue. Therefore, lack of recognition and misdiagnosis are likely contributing factors leading to underdiagnosis and -treatment of dVIN patients and subsequent progression towards VSCC.

## 5. Conclusions

The results of this review confirm the clinical impression that the risk of developing primary and recurrent VSCC in dVIN is high with a short time to cancer progression. Adequate treatment by radical excision and careful surveillance of patients with dVIN, preferably in a gynecologic oncology center with multidisciplinary expertise, is therefore highly recommended. Patients ought to be well informed on the high cancer risk and the need for frequent visits with regular biopsies, especially in the presence of symptoms. Rapid progression to VSCC and lack of recognition cause dVIN to be a rare finding. Novel techniques and additional biomarkers are currently being investigated to aid in diagnostics of this rare and aggressive disease. Prospective studies with larger well-defined cohorts, preferably population-based, are highly needed to further investigate the cancer risk in dVIN.

## Figures and Tables

**Figure 1 cancers-13-06170-f001:**
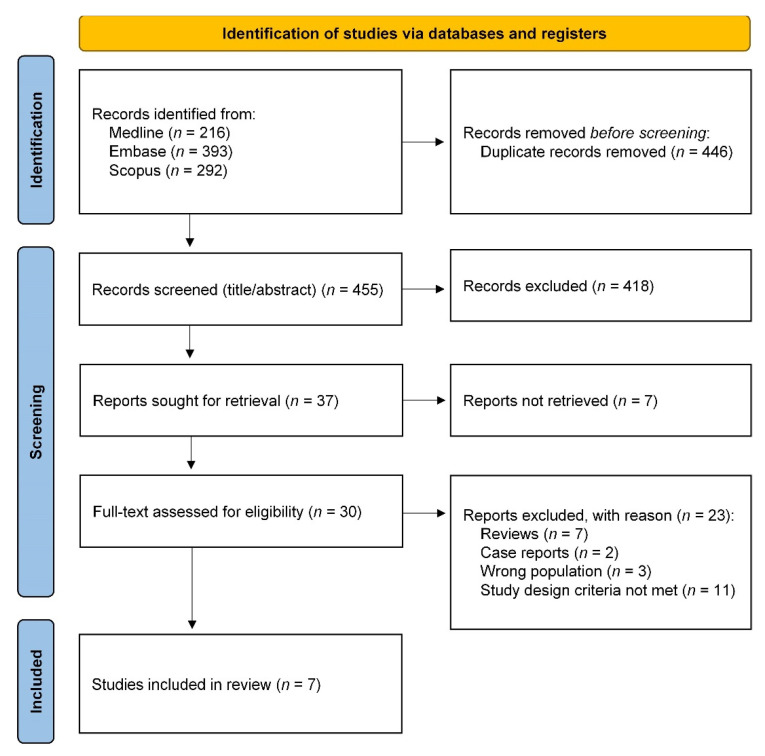
Preferred Reporting Items for Systematic Reviews and Meta-Analyses (PRISMA) flow diagram for selection of studies [15].

**Table 1 cancers-13-06170-t001:** Characteristics of included studies.

Study (Year)	Country	Study Design	Type of Cohort	Inclusion Period	Number and Type of Cases	Inclusion Criteria dVIN
Yang et al. (2000) [22]	USA	Retrospective cohort study	Center-based	NA	8 dVIN ^1^	Revision of pathology slides
Van de Nieuwenhof et al. (2009) [18]	The Netherlands	Retrospective cohort study	Population-based	1992–2005	67 dVIN	Pathology reports describing dVIN as differentiated VIN, VIN simplex type, VIN NOS with LS and/or a high-risk HPV-negative result
Regauer et al. (2016) [19]	Austria	Retrospective cohort study	Center-based	2004–2016	16 dVIN	Revision of pathology slides
Thuijs et al. (2020) [7]	The Netherlands	Retrospective cohort study	Population-based	1991–2011	12 dVIN	Pathology reports describing dVIN as differentiated VIN, vulvar dystrophy with atypia or simplex VIN
McAlpine et al. (2017) [20]	Canada	Retrospective cohort study	Center-based	1985–2005	7 dVIN18 dVIN/VSCC	Revision of pathology slides
Eva et al. (2008) [21]	UK	Retrospective cohort study	Center-based	2000–2005	44 dVIN/VSCC	Pathology reports describing dVIN adjacent to VSCC
Te Grootenhuis et al. (2019) [17]	The Netherlands	Retrospective cohort study	Center-based	2000–2010	197 dVIN/VSCC	Revision of pathology slides

^1^ Referred to as simplex VIN in the article. Abbreviations: dVIN, differentiated vulvar intraepithelial neoplasia (cases without history of vulvar cancer); dVIN/VSCC, dVIN adjacent to vulvar squamous cell carcinoma; HPV, human papillomavirus; LS, lichen sclerosus; NA, not available; NOS, not otherwise specified; UK, United Kingdom; USA, United States of America; VIN, vulvar intraepithelial neoplasia.

**Table 2 cancers-13-06170-t002:** Risk of bias summary: judgement of each domain for all included studies using the Quality of Prognostic Studies (QUIPS) tool [16].

Study (Year)	Study Participation	Study Attrition	Prognostic Factor Measurement	Outcome Measurement	Study Confounding	Statistical Analysis and Reporting
Yang et al. (2000) [22]	High	High	Moderate	Low	Moderate	Moderate
Van de Nieuwenhof et al. (2009) [18]	Low	Low	Moderate	Low	High	Low
Regauer et al. (2016) [19]	High	High	Moderate	High	High	High
Thuijs et al. (2020) [7]	Low	Low	Moderate	Low	High	Low
McAlpine et al. (2017) [20]	Low	Moderate	Low	Low	Moderate	Low
Eva et al. (2008) [21]	Moderate	High	Moderate	High	High	Moderate
Te Grootenhuis et al. (2019) [17]	Low	Moderate	Low	Low	Moderate	Low

For all included studies, risk of bias was scored as low risk of bias (depicted in green), moderate risk of bias (depicted in yellow) or high risk of bias (depicted in red) for each of the six domains.

**Table 3 cancers-13-06170-t003:** Outcome of objectives of all included studies.

Study (Year)	Number and Type of Cases	Age	Follow-Up Time	Primary VSCC Risk in dVIN			Recurrent VSCC Risk in dVIN/VSCC		
				Risk of Primary VSCC		Time to Progression	Risk of Recurrent VSCC		Time to Progression
		Years, Median (Range)	Months, Median (Range)	Number of Primary VSCC	Absolute Risk (%)	Months, Median (Range)	Number of Recurrent VSCC	Absolute Risk (%)	Months, Median (Range)
Yang et al. (2000) [22]	8 dVIN ^1^	67.5 (55–82)	48.5 (14–121)	3	37.5	9 (6–55)	-	-	-
Van de Nieuwenhof et al. (2009) [18]	67 dVIN	67	NA	22	32.8	22.8 (3–84)	-	-	-
Regauer et al. (2016) [19]	16 dVIN	NA	NA	9	56.3	NA	-	-	-
Thuijs et al. (2020) [7]	12 dVIN	70.3 (40.3–85.3)	167 (4–329) ^2^	7	58.3	16.8	-	-	-
McAlpine et al. (2017) [20]	7 dVIN 18 dVIN/VSCC	75.1 ^3^ 75.8 ^3^	72 NA	6 -	85.7 -	22.8 -	- 17	- 94.4	- 13.2
Eva et al. (2008) [21]	44 dVIN/VSCC	NA	NA	-	-	-	14	31.8	NA
Te Grootenhuis et al. (2019) [17]	197 dVIN/VSCC	73 (26–100) ^4^	80 (0–204)	-	-	-	94 ^5^	47.7	32 (0–202) ^6^

^1^ Referred to as simplex VIN in the article; ^2^ Follow-up time for entire study population (*n* = 894); ^3^ Mean age; ^4^ Median age for entire study population (*n* = 287); ^5^ Number of cases calculated from reported 10-year recurrent risk in dVIN/VSCC cohort; ^6^ Time to progression for entire study population (*n* = 287), did not differ significantly between adjacent precursor lesion groups (*p* = 0.08). Abbreviations: dVIN, differentiated vulvar intraepithelial neoplasia (cases without history of vulvar cancer); dVIN/VSCC, dVIN adjacent to vulvar squamous cell carcinoma; NA, not available; VSCC, vulvar squamous cell carcinoma.

## Data Availability

Data are available upon request from the authors.

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
