# Peer review of "The Vulvar Cancer Risk in Differentiated Vulvar Intraepithelial Neoplasia: A Systematic Review"

_cancers, 2021, doi:10.3390/cancers13246170_

Round 1

Reviewer 1 Report

The aim of the authors in this study was to systematically review the literature on the risk of primary and recurrent vulvar squamous cell carcinoma in patients with differentiated vulvar intraepithelial neoplasia, taking into account the time to tumor progression.

Due to the rare occurrence of 26 dVIN (Differentiated vulvar intraepithelial neoplasia), there is limited information in the literature on the exact risk of developing this type of cancer. The authors' results underscore the importance of promptly identifying and treating differentiated vulvar intraepithelial neoplasia for the prevention of vulvar cancer progression.

The authors' results underscore the importance of promptly identifying and treating differentiated vulvar intraepithelial neoplasia for the prevention of vulvar cancer progression.

My questions:

“The absolute risk for primary VSCC in dVIN varied between 33 and 86%, with a median time to progression to VSCC of 9 – 23 months.

The risk of developing recurrent 33 VSCC in dVIN associated VSCC was 32 – 94%, with a median time to recurrence of 13 – 32 months.”

whether the given results were counted by the authors or analyzed on the basis of selected publications. If it was counted by the authors, how it was done (what program was used for it) and if there are charts or drawings available, then they should be included in the paper or supplement.

Author Response

We thank the reviewer for the comment and agree that this can be clarified. Depending on the article, the given results (primary and recurrent VSCC risk and time to progression to primary and recurrent VSCC) were extracted either directly from the text or calculated from the data presented in the text and tables. Extracted results were collected in Microsoft Excel.

Absolute risk was calculated by dividing the number of cases developing primary/recurrent VSCC by the number of dVIN or dVIN adjacent to VSCC cases. For one article (Te Grootenhuis et al.) we calculated the absolute risk for recurrent VSCC from the reported 10-year recurrent risk, which is reported in the footnote of table 3 on page 8. Median time to progression was reported in the text or calculated from provided data. All calculations were performed in Microsoft Excel. Additional charts or drawings were not created.

To clarify the above we added information in section 2.3: Data extraction and analysis on page 3. The following adjustments were made:

Page 3, line 178-179: Data from selected studies were extracted from the full text or tables by four reviewers (MCGB, RFMV, NBT, FOV).

Page 3, line 185-188: Absolute risk was calculated by dividing the number of cases developing primary and/or recurrent VSCC by the number of dVIN and/or dVIN adjacent to VSCC cases. Data were analyzed in a descriptive manner if meta-analysis could not be performed. The data were tabulated in Microsoft Excel version 2016.

Reviewer 2 Report

this is an article on an important topic the frequency with which the dVIN turns into SCC. I believe that dVINs do not go under valued.
Some little hints: Put the colors on table 2 with low being red, moderate being yellow and high being green. And then say something about the elderly if it is more frequent or if you can tell the difficulty of the diagnosis

Author Response

Thank you for your suggestions. We agree that including colors will make Table 2 more clear for the reader. Studies with a high risk of bias were colored red, moderate risk of bias colored yellow and low risk of bias colored in green. An adjusted table has been included in the manuscript on page 7.

We do agree that it is important to highlight attention on the elderly and the diagnostic difficulties. In the introduction we mention that dVIN occurs mostly in older women above the age of 60 (page 2, lines 131-133). To further emphasize the higher frequency of dVIN in older women we added a sentence to the discussion on page 10:

Page 10, lines 305-307: Both studies observed that dVIN occurs more frequently in older women, with a peak incidence amongst women between age 75 and 85 years.

In the discussion we extensively elaborate on the difficulty of the diagnosis (page 11, lines 362-395). To further highlight this point we also added the following sentence to the introduction on page 2:

Page 2, lines 139-140: Solitary dVIN is a rare finding and difficult to diagnose due to its varying presentation.

Reviewer 3 Report

This is a very well written systematic review summarizing the risk of development and recurrent invasive squamous cell carcinoma in the setting of differentiated vulvar intraepithelial neoplasia. This is a relatively rare entity and there are very few comprehensive studies. The authors have included pertinent studies and have discussed the limitations including sample size, nomenclature, histopathology review (or lack thereof), etc.  The results are very well organized, it is comprehensive and easy to read.

My only minor suggestion would be on page 10, lines 272-276, the authors briefly mention DE-VIL and VAAD. These are relatively recently described entities that appropriately belong in the differential diagnosis of dVIN. There are a few recent publications, including one by Dr. Parra-Herran in Modern Pathology, covering molecular findings, that may be worth mentioning.

Author Response

Thank you for your comments. In response to your suggestion we added a sentence to the discussion on page 10, in which we added a reference of the recent publication by Dr. Parra-Herran (reference number 31) on the molecular association of DEVIL/VAAD and verrucous carcinoma:

Page 11, line 367-369: These entities show overlapping morphology with dVIN and previous studies suggest that a molecular association exists with verrucous carcinoma, a rare variant of HPV-independent VSCC [29,31].

Page 15, line 528-531: 31. Akbari, A.; Pinto, A.; Amemiya, Y.; Seth, A.; Mirkovic, J.; Parra-Herran, C. Differentiated exophytic vulvar intraepithelial lesion: Clinicopathologic and molecular analysis documenting its relationship with verrucous carcinoma of the vulva. Mod Pathol 2020, 33, 2011-2018, doi:10.1038/s41379-020-0573-5.